# Telemonitoring at scale for hypertension in primary care: An implementation study

**Vicky Hammersley**[1], **Richard Parker**[1], **Mary Paterson**[1], **Janet Hanley**[2], **Hilary Pinnock**[1], **Paul Padfield**[1], **Andrew Stoddart**[1], **Hyeon Gyeong Park**[1], **Aziz Sheikh**[1], **Brian McKinstry**[1]*

**1** Usher Institute, University of Edinburgh, Edinburgh, United Kingdom, **2** School of Health and Social Care. Edinburgh Napier University, Edinburgh, United Kingdom

* brian.mckinstry@ed.ac.uk

## Abstract

### Background

While evidence from randomised controlled trials shows that telemonitoring for hypertension is associated with improved blood pressure (BP) control, healthcare systems have been slow to implement it, partly because of inadequate integration with existing clinical practices and electronic records. Neither is it clear if trial findings will be replicated in routine clinical practice at scale. We aimed to explore the feasibility and impact of implementing an integrated telemonitoring system for hypertension into routine primary care.

### Methods and findings

This was a quasi-experimental implementation study with embedded qualitative process evaluation set in primary care in Lothian, Scotland. We described the overall uptake of telemonitoring and uptake in a subgroup of representative practices, used routinely acquired data for a records-based controlled before-and-after study, and collected qualitative data from staff and patient interviews and practice observation. The main outcome measures were intervention uptake, change in BP, change in clinician appointment use, and participants' views on features that facilitated or impeded uptake of the intervention. Seventy-five primary care practices enrolled 3,200 patients with established hypertension. In an evaluation subgroup of 8 practices (905 patients of whom 427 [47%] were female and with median age of 64 years [IQR 56–70, range 22–89] and median Scottish Index of Multiple Deprivation 2012 decile of 8 [IQR 6–10]), mean systolic BP fell by 6.55 mm Hg (SD 15.17), and mean diastolic BP by 4.23 mm Hg (SD 8.68). Compared with the previous year, participating patients made 19% fewer face-to-face appointments, compared with 11% fewer in patients with hypertension who were not telemonitoring. Total consultation time for participants fell by 15.4 minutes (SD 68.4), compared with 5.5 minutes (SD 84.4) in non-telemonitored patients. The convenience of remote collection of BP readings and integration of these readings into routine clinical care was crucial to the success of the implementation. Limitations include the fact that practices and patient participants were self-selected, and younger and

**Data Availability Statement:** Scale-up BP made use of several routine electronic health care data sources that are linked, de-identified, and held in the NHS Lothian safe haven, which is only

accessible by approved individuals who have undertaken the necessary governance training. Therefore, raw participant-level data is not available for public sharing. Requests to access data should first be made to NHS Lothian Safe Haven (https://www.accord.scot/researcher-access-research-data-nrs-safe-haven/safe-haven-network) who will assist with further access for researchers who meet the criteria for access to confidential data.

**Funding:** BM, JH, RP, HP, PP,AS were supported by a grant the Chief Scientist Office of the Scottish Government CZH/4/1135. This funding source had no role in study design, data collection, data analysis, data interpretation, writing of the report, or the decision to submit the paper for publication. The authors had full access to all the data in the study and had final responsibility for the decision to submit for publication.

**Competing interests:** I have read the journal's policy and the authors of this manuscript have the following competing interests: BM is supported by the Scottish Government in relation to their plans to scale up telemonitoring for hypertension across Scotland. MP is paid by the Scottish Government to give advice on implementing telemonitoring of blood pressure. BM and ASh are in receipt of funding for an unrelated hypertension telemonitoring study of people with stroke. ASh is a member of the Editorial Board of PLOS Medicine. ASt has received research funding for this study and another trial of Telehealth for Blood Pressure. HP has received fundng in the last 3 years from the European EIT Digital fund to develop an app for BP management. All other authors declare no support from any organisation for the submitted work; no financial relationships with any organisations that might have an interest in the submitted work in the previous three years; no other relationships or activities that could appear to have influenced the submitted work.

**Abbreviations:** BP, blood pressure; DDD, defined daily dose; GP, general practitioner; RCT, randomised controlled trial; SIMD, Scottish Index of Multiple Deprivation.

more affluent than non-participating patients, and the possibility that regression to the mean may have contributed to the reduction in BP. Routinely acquired data are limited in terms of completeness and accuracy.

## Conclusions

Telemonitoring for hypertension can be implemented into routine primary care at scale with little impact on clinician workload and results in reductions in BP similar to those in large UK trials. Integrating the telemonitoring readings into routine data handling was crucial to the success of this initiative.

## Author summary

### Why was this study done?

- Although uncontrolled hypertension is the biggest remediable cause of stroke and myocardial infarction, and anti-hypertensive medications are effective, many patients have uncontrolled blood pressure (BP).

- Despite strong evidence that telemonitoring encourages medication use and is effective at lowering BP, clinicians have been slow to adopt it, in part due to poor integration with routine clinical processes.

- We wanted to see if an integrated telemonitoring system would be taken up by primary care clinicians at scale, what impact this would have on their workload, and if changes in BP would match those of randomised controlled trials.

### What did the researchers do and find?

- Based on our previous research with clinicians on the desirable attributes of a telemonitoring system, NHS Lothian developed an integrated system that provided regular summaries of patient home-monitored BP readings to their general practitioner, which were delivered alongside routine laboratory results.

- We observed the rollout of this system, interviewing patients and clinicians about their attitudes towards innovations to determine what worked and did not in terms increasing uptake.

- In a group of 8 practices, we collected routinely acquired data on BP, clinician appointments, and other resource use and compared these to the previous year. Resource use was compared with patients in these practices with high BP who did not use telemonitoring.

- We found that the intervention was popular in many but not all practices. Patients who used the system had fewer appointments in the year of the intervention compared with the previous year. BP fell in the intervention group, in line with findings of UK randomised controlled trials.

**What do these findings mean?**

- The findings suggest that introducing telemonitoring to routine practice at scale is feasible.

- Although not definitive, the findings provide some reassurance that the intervention did not increase practice workload and that in routine practice the improvements in BP control were similar to those in controlled trials.

- However, the people who took part were not entirely typical of practice populations as a whole, being on average younger and slightly more affluent.

- The findings support plans to introduce telemonitoring more generally, but within an evaluative framework.

## Introduction

Hypertension is common among people over the age of 50 years and is an important risk factor for cardiovascular disease [1]. Although effective management greatly reduces the risk of cardiovascular events, blood pressure (BP) remains uncontrolled in many people [2]. This is in part due to poor adherence to medication [3], but also reluctance on the part of clinicians to intensify therapy [4] and on the part of patients to accept intensified therapy [5].

In the UK, hypertension is managed in primary care mainly by practice nurses assisted by healthcare assistants with specific training in BP monitoring, supported by general practitioners (GPs). In Scotland (population 5.45 million), 1.2 million appointments in 2018 were taken up solely for hypertension checks [6], despite evidence that home monitoring is a better predictor of long-term outcomes than office measurement [7]. Under the Scottish National Health Service, all primary care attendances and medications are free at the point of contact, paid for from general taxation. However, primary care is currently under considerable stress, with falling numbers of GPs [8] and rising workload [9] as populations age and the prevalence of hypertension increases. Technology has been promoted as a strategy for managing this workload [10,11].

A number of trials have demonstrated that patient self-monitoring has a small but statistically significant effect on improving BP control [12]. Telemonitoring, however, which engages clinicians in reviewing readings taken by patients and submitted over the internet or via short message service (SMS), results in much larger clinically significant reductions in BP [13] and is cost-effective [14,15]. Telemonitoring is associated with an intensification of medication [16], and qualitative research has shown that this is because multiple persistently raised BP readings convince both clinicians and patients that medication intensification and improved adherence is required, thus overcoming 'therapeutic inertia' [5]. Supported self-monitoring enables patients to act on physiological measures rather than on (perceived) symptoms such as headache or fatigue [17].

Why then, given consistent findings from randomised controlled trials (RCTs), has telemonitoring not been more widely adopted? Implementing new models of care at scale is challenging [18,19], particularly in the context of clinical teams already working at full stretch. Telehealth trials have tended to recruit relatively few, often highly selected individuals who have then been followed up for relatively short periods, leaving unanswered questions about the day-to-day practicality of managing large numbers of patients [20]. In

addition, the adoption, effectiveness, and impact on resources of delivering BP telemonitoring as a routine approach to care are unknown and may differ from the experience in trials.

Following our earlier RCT of telemonitoring for uncontrolled hypertension [16], we sought to investigate the acceptability and impact of introducing long-term telemonitoring across NHS Lothian in southeast Scotland for people with previously diagnosed hypertension.

## Methods

This study is reported in accordance with the Standards for Reporting Implementation Studies (StaRI) [21]. This study was approved by the East of England–Cambridge South Research Ethics Committee (16/EE/0058). We describe the deployment and uptake of telemonitoring generally and then the evaluation of the impact of the implementation in 8 practices chosen to represent a range of size, deprivation, and earlier and later adopter of the technology.

### Setting

The study was carried out in Lothian, Scotland (population 858,000). This region of Scotland, which contains Edinburgh, the capital of Scotland, is mainly urban, with a mix of small towns, suburban areas, inner city, and some rural areas. There are a range of levels of affluence and deprivation.

### The Scale-Up BP telemonitoring system and deployment

**Logic model.** The logic model for the Scale-Up BP deployment is illustrated in Fig 1, with primary (implementation and health) outcomes and process outcomes mapped to the mechanism by which the Scale-Up BP implementation strategy and telemonitoring was anticipated to work.

The telemonitoring intervention implemented is presented in detail in S1 Text and S1 Fig. Participating people with hypertension were provided with an electronic oscillometric sphygmomanometer and shown how to submit BP readings via a low-cost third-party text-based telemonitoring system (Florence) [22] using their own mobile phone. These readings were

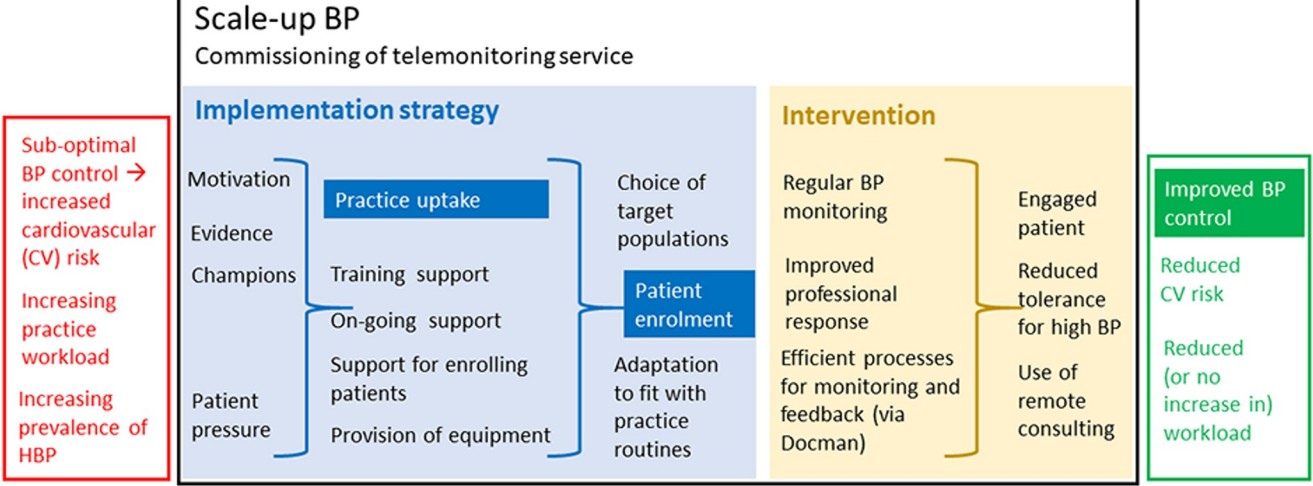

**Fig 1. This logic pathway illustrates how the Scale-Up BP implementation strategy was expected to work.** The primary outcomes are in white font on a dark blue (implementation outcomes) or dark green (health outcome) background. The distinction is made between the evidence-based intervention (BP telemonitoring) (yellow field) and the Scale-Up BP implementation strategy used to implement BP telemonitoring in routine practice (blue field). BP, blood pressure; HBP, high blood pressure.

stored in a central server and made available to practices via an internet link. However, previous research had shown that a major barrier to adoption was the need for clinicians to log on to third-party websites during busy clinics [20], and so a novel element [23] was developed that automatically extracted patient-generated data from the third-party website. This element displayed a mean BP and summarised BP values in graphical and tabular format (S2 Fig) and dispatched these summaries, at intervals chosen by the clinician, through the routinely used primary care data management system, Docman [24]. This obviated the need for third-party log-ons and, critically, systematically presented the telemonitoring results in manageable numbers on a daily basis, with relatively infrequent but data-rich reports integrated into the electronic medical record and seen alongside routine laboratory results and hospital communications. Between reports, patients were informed by automated text responses if submitted readings were low, normal, high, or very high and were advised to follow a written action plan with respect to contacting their practice either routinely or urgently as appropriate.

The Scale-Up BP implementation project was launched in June 2015 and continues. The Scottish Government's Technology Enabled Care (TEC) Programme [25] financed the third-party telemonitoring service, the development of the software to link it with primary care systems using Docman, supported facilitators to visit/train practices, and purchased sphygmomanometers for loan to patients.

**Recruitment of practices.** All 126 Lothian practices were invited to participate via a weekly newsletter. Practices expressing interest were followed up with information visits and training if they participated.

## The Scale-Up BP implementation strategy

The implementation strategy was informed by barriers and facilitators identified in previous research as important to clinicians [20,26] and utilised elements of 'COM-B' behaviour change [27] and diffusion of innovation theories [28]. S1 Table lists how these barriers were addressed. In essence the approach comprised the following: motivating clinicians and service planners by demonstrating how an evidence-based BP telemonitoring service could improve outcomes and potentially save time; enhancing capability by providing initial training and (importantly) retraining and ongoing support as required; and providing opportunity by supporting recruitment drives within individual practices. Local practitioner 'champions' were recruited to be early adopters in the expectation that they would demonstrate and promote the intervention to colleagues [29]. Practices were encouraged to adapt the system to suit their routines and priorities, with flexibility in terms of the initial patient groups targeted (some focusing on uncontrolled hypertension). Clinicians were continuously involved in the development of the implementation strategy, trialling different recruitment methods and different approaches to managing telemonitoring data. Learning from practices was shared in regular newsletters.

## Outcomes of interest

Our primary implementation outcomes were the number of participating practices and the overall number of patients recruited. In addition, in a subgroup of 8 practices, chosen to represent a range of socioeconomic status, practice size, city centre/small town, and fast/slow patient recruitment, we measured patient engagement and adherence, resource and medication use, and BP change over time, and gauged clinician and patient acceptability of the implementation.

These practices agreed to allow data on BP, anti-hypertensive medication prescriptions, and practice workload (as recorded in the appointment system) for all patients with hypertension (including those not being managed by the Scale-Up BP system) to be extracted from

their electronic health record, linked to data from the telemonitoring system, de-identified, and transferred to the local NHS safe haven (a secure analysis server that does not permit any data to be taken away).

This allowed comparison of resource use between participating and non-participating patients. Primary care stored data were collected and transferred by a data extraction service (Albasoft) through software already resident on the practice computers. Time to action for very high (>160 mm Hg) and very low (<90 mm Hg) average systolic BP readings was obtained from manual searching of Docman reports and practice records. Change in BP over time was derived from telemonitored data from Scale-Up BP patients only.

As the purpose of the study was primarily to determine uptake of the intervention, a formal power calculation was not deemed appropriate.

## Assessment of attrition

Patients were considered to have discontinued monitoring if they did not record any readings for 7 months during the observation period. This period was chosen because we did not want to exclude patients with good BP control, some of whom were asked to submit readings only every 6 months.

A logistic mixed regression analysis was performed on the discontinuation outcome to help us identify the characteristics of patients who had discontinued. The variables female sex, second telemonitored systolic BP, Scottish Index of Multiple Deprivation (SIMD) [30] $\geq 5$ (less deprived), and age were included in the model, which was also adjusted for GP practice as a random effect. The continuous variables systolic BP and age were initially fitted as having a continuous (linear) relationship with the probability of discontinuation, but this assumption was subsequently relaxed in sensitivity analysis, whereby we used flexible natural B-spline functions to model the relationship between these variables and the probability of discontinuation. The modelled relationships were then displayed graphically in line plots.

## Practice workload and resource use

We collected data on (i) the total number of appointments, (ii) total number of face-to-face appointments, (iii) total consultation time, and (iv) total consultation time in the surgery.

The practice appointment system was used to identify all interactions including face-to-face and planned telephone consultations, home visits, and administration activities (for example, prescribing a requested repeat prescription). Face-to-face consultations could be reliably identified as they had different arrival and start times. We determined time spent in activities recorded in the appointment system in total and in face-to-face activities for all people with hypertension in the 8 practices and could compare those taking part in the intervention with those who were not.

There was no automated extraction method that could determine time spent in consultations that were not recorded in the appointment system such as ad hoc phone calls or record checking.

Analysis of resource use outcomes was conducted by computing totals/averages across all patients and comparing before and after Scale-Up BP implementation, and separately performing a patient-level analysis by computing sum totals/averages within patients. We only included patients with a full year of data before and after the first recorded telemonitored reading. Regarding the consultation time analysis, the first analysis weighted patients with a greater number of appointments more heavily than the second patient-level analysis did, which is appropriate when simply trying to give an overall impression of resource use burden. On the other hand, the second analysis takes a within-patient approach to ascertain if consultation

times have changed at the patient level, after properly accounting for clustering by patient. Both sets of analyses are complementary in allowing us to provide a full and clear presentation of the resource use data.

The telemonitored patients' appointment data were compared against comparator patients (all patients with a diagnosis of hypertension who did not use the telemonitoring system) from the same 8 evaluation practices. For the comparator group, we could not choose the start of telemonitoring as the 'anchor point' to define the before and after groups, we therefore used a randomly chosen date for the anchor point within a date range consistent with patient telemonitoring start times in the telemonitoring group (between October 2015 and June 2016). Outlier consultation durations (>30 minutes, which is 3 times the length of a typical primary consultation in the UK) were excluded from all analyses involving consultation time as these are very likely to be erroneous, and so it was not appropriate to retain them in the analyses. These long consultation times are almost certainly because clinicians failed to close the electronic record, for example before taking breaks or starting other tasks. It would be very unlikely that a consultation related to BP management would be so long; including these would lead to an overestimate of resource use.

Linear mixed effects models were then fitted to the resource use outcomes to compare telemonitored patients with the comparator group, adjusting for female sex, SIMD $\geq 5$ (less deprived), initial surgery-measured systolic BP (after September 2015), age (in years), and GP practice as a random effect. As was the case for the attrition analysis, natural B-spline functions were used to model the effects of age and systolic BP on the outcome.

## Change in BP over 6 months and proportion of patients with controlled BP

The difference between the second telemonitored BP reading (baseline) and the last BP reading, occurring 6–12 months after the second telemonitored reading, was calculated for each patient (second minus last). We used the second telemonitored reading because the first was sometimes taken in the GP practice as a demonstration of the system to the patient, and there is a known disparity between home and office BP readings. Only patients with a full year's follow-up were included. Summary statistics (mean, median, standard deviation, lower quartile, upper quartile, minimum, and maximum) were calculated based on these BP differences, overall and stratified by age, sex, SIMD, and second telemonitored BP.

To assess the proportion of patients whose hypertension was uncontrolled, as defined by National Institute for Health and Care Excellence guidelines [31] (home systolic BP $\geq 135$ mm Hg), we categorised the BP data according to level of control of BP, and compared the proportions with uncontrolled hypertension at baseline with the proportion 6–12 months later. Patients were only included in this analysis if they had complete data at baseline and 6–12 months.

## Assessment of medication prescribed

For each patient, we calculated the total number of anti-hypertensive medication prescriptions in the year before the patient first started telemonitoring and in the year after. To avoid bias, we only included patients with a full year of data before and after the start of telemonitoring. Summary statistics were then computed overall and stratified by age, sex, SIMD, and initial systolic BP. SAS software version 9.4 (SAS Institute, Cary, NC, US) was used to analyse the data.

Mean defined daily dose (DDD) was calculated for the 6 months before and after date of the first telemonitored reading for all patients with at least 6 months of follow-up, based on

WHO DDD index values [32], and compared using a paired *t* test (analysis undertaken in R version 3.5.3).

## Qualitative exploration of patient and professional perceptions

In the 8 evaluation practices, we undertook semi-structured face-to-face interviews with patients, GPs, practice nurses, and healthcare assistants, to explore perceptions of the telemonitoring intervention in terms of ease of use, organisational adoption and adaptation, barriers, facilitators, and potential improvements to the implementation of Scale-Up BP. See S2 Text for details of methodology.

# Results

## Progress of the implementation: Practice and patient recruitment

The number of participating practices throughout Lothian grew steadily following the launch, and by July 2019 (45 months after the start of the project) 75/126 practices in Lothian had participated. Patient recruitment accelerated as new practices were trained, reaching a total of 3,200 patients by July 2019 (S3 Fig). Practices differed greatly in terms of the numbers of patients they recruited (range 6–400). Some required several training sessions before commencing recruitment. The most successful recruiting practices systematically invited all their patients with hypertension to register. Two practices had evening group meetings. Others, with highly motivated clinical staff, were very successful at recruiting patients during routine attendances. In some practices, clinicians were selective about to whom they offered the intervention. For example, initially, when they had less experience with the system, they tended to offer it to patients whom they thought would be most likely to manage self-monitoring. Some practices focused initially on uncomplicated patients with controlled BP, while others chose patients with less well controlled BP.

## Patient population

In the 8 evaluation practices, of the 905 patients signed up to use the telemonitoring system and providing BP data, 427 were female (47%), and the patients had a median age of 64 years (IQR 56–70, range 22–89) and median SIMD 2012 decile of 8 (IQR 6–10). The individuals in the comparator group were older on average (median age 71 years, IQR 62–79, range 20–90), had a slightly higher percentage of females (54%), and were slightly less affluent on average (median SIMD of 7). Initial systolic BP reading was similar between groups (see Table 1). The median number of patients using the telemonitoring system per practice was 80 (IQR 44 to 153), compared to 450 (IQR 332 to 599) controls. Apart from being slightly more affluent, generally the evaluation practices were representative of practices in Lothian (see Table 2).

**Table 1. Patient characteristics in the telemonitoring and comparator groups.**

| Characteristic | Telemonitoring group (*n* = 905) | Comparator group (*n* = 9,061) |
|---|---|---|
| Female sex | 427/905 (47%) | 4,934/9,061 (54%) |
| Age of patient in 2015 (years)* | Median 64 (IQR 56 to 70, min 22, max 89, *n* = 905) | Median 69 (IQR 60 to 79, min 19, max 90, *n* = 8,610) |
| SIMD 2012 decile | Median 8 (IQR 6 to 10, min 2, max 10, *n* = 888) | Median 7 (IQR 5 to 10, min 1, max 10, *n* = 8,957) |
| Initial systolic blood pressure reading in the surgery (after September 2015) | Median 140 (IQR 130 to 150, min 90, max 200, *n* = 877) | Median 138 (IQR 129 to 149, min 71, max 240, *n* = 7,694) |

*The exact date of birth was not available because of data governance reasons.

max, maximum; min, minimum; SIMD, Scottish Index of Multiple Deprivation.

**Table 2. Descriptive statistics for evaluation practices compared to non-evaluation practices in NHS Lothian.**

| Characteristic | Non-evaluation practices (*n* = 117) | Evaluation practices (*n* = 8) |
|---|---|---|
| Percentage female | 50.7% (50.0% to 51.5%) | 51.0% (50.7 to 51.5%) |
| Percentage 25–44 years old | 28.5% (25.3% to 36.1%) | 27.1% (25.6% to 36.7%) |
| Percentage 45–64 years old | 27.0% (24.2% to 28.6%) | 27.6% (18.6% to 29.2%) |
| Percentage 65+ years old | 16.0% (12.7% to 19.5%) | 17.3% (10.6% to 20.6%) |
| Percentage SIMD < 5 (more deprived) | 31.5% (12.2% to 50.7%) | 22.9% (12.1% to 38.3%) |
| Modal urban/rural classification | Large urban areas, 68 (59%)<br>Other urban areas, 23 (20%)<br>Accessible small towns, 12 (10%)<br>Accessible rural, 8 (7%)<br>Remote small towns, 5 (4%) | Large urban areas, 5 (62%)<br>Other urban areas, 2 (25%)<br>Accessible small towns, 1 (12%)<br>Accessible rural, 0 (0%)<br>Remote small towns, 0 (0%) |

Results are expressed as median (interquartile range), except modal urban/rural classification, which is presented as frequency (percentage).

SIMD, Scottish Index of Multiple Deprivation.

## Patient engagement

We explored patient engagement during the period 2 September 2015 to 7 January 2018 in the 8 evaluation practices. Of 905 patients signed up to use the telemonitoring system and providing BP data, a median of 28 (IQR 14 to 76) BP readings were submitted in total per patient, although many of these patients were only recently recruited and had varying lengths of follow-up. Excluding patients with less than 1 year of follow-up, the remaining 430 patients submitted a median of 42 readings in 1 year (IQR 22 to 94), with 359/430 patients (83%) submitting over 20 readings and 97/430 patients (23%) submitting over 100 readings (see S4 Fig).

## Patient attrition

Of the 655 patients who had been enrolled in telemonitoring for more than 7 months, 49 (7%) stopped texting readings. Sex, age, or SIMD did not significantly predict discontinuing (see S2 Table). However, patients with higher systolic BP were more likely on average to discontinue (odds ratio 1.03, 95% CI 1.01 to 1.05). S5 and S6 Figs show how the predicted probability of dropout varies with systolic BP and age when these variables are fitted using flexible spline functions and are not constrained to be linear in the statistical model.

## Change in resource use over time

According to records in the appointment system, there were 1,260 appointments in total (671 face-to-face) in the year before the start of telemonitoring, compared to 1,158 appointments in total (569 face-to-face) in the year after. This corresponded to an observed 8% reduction in total appointments, and a 15% reduction in the number of face-to-face appointments.

There was an observed increase in appointments just prior to the start of telemonitoring, probably due to patients attending for training to measure BP and text results, which might potentially exaggerate potential reductions in consultation numbers (see S7 Fig). In further analysis, in comparing changes in resource use with a comparator group of non-participating patients, we therefore excluded consultations within 2 weeks of the anchor point, which for telemonitoring patients may have involved appointments to set up the system. We also excluded 7% of consultations for which the appointment times were not recorded and an additional 3% of consultations that were recorded as being over 30 minutes in duration.

**Table 3. Change in total and face-to-face appointment numbers and consultation length in people with hypertension participating and not participating in telemonitoring.**

| Outcome | Variable | Telemonitoring group (*n* = 118) | | | Comparator group (*n* = 9,061) | | |
|---|---|---|---|---|---|---|---|
| | | Median | Lower quartile | Upper quartile | Median | Lower quartile | Upper quartile |
| Number of appointment activities per patient | Before | 6.5 | 4 | 14 | 7 | 4 | 14 |
| | After | 6 | 4 | 13 | 7 | 3 | 14 |
| | Reduction (before minus after) | 0 | −3 | 4 | 0 | −4 | 4 |
| Number of face-to-face consultations per patient | Before | 5 | 3 | 9 | 4 | 1 | 8 |
| | After | 4 | 2 | 7 | 4 | 1 | 7 |
| | Reduction (before minus after) | 1 | −2 | 3 | 0 | −2 | 3 |
| Consultation time per patient all appointments (minutes) | Before | 67.5 | 30 | 129 | 64 | 27 | 123 |
| | After | 49.5 | 25 | 103 | 59 | 23 | 116 |
| | Reduction (before minus after) | 14 | −25 | 53 | 4 | −32 | 39 |
| Face-to-face consultation time per patient (minutes) | Before | 63 | 27 | 104 | 47 | 15 | 97 |
| | After | 41 | 19 | 82 | 42 | 10 | 88 |
| | Reduction (before minus after) | 15 | −18 | 48 | 0 | −23 | 34 |

Both the comparator group and telemonitoring group showed decreases in numbers of all appointments and face-to-face appointments. Decreases in the number of appointments were notably more pronounced in the telemonitored group than in the comparator group for females, relatively more deprived (SIMD < 5) individuals, and people whose BP was controlled (see S3 Table).

Table 3 shows the number of consultations and consultation length for the participant and comparator groups for the year before and after the intervention. A total of 118 telemonitoring patients were included with at least 1 consultation recorded before and after the intervention. As above, we excluded consultations within 2 weeks of the anchor point. Among these 118 patients, 46 (39%) were female, and they had a median age of 64 years (IQR 54 to 69), median SIMD of 9 (IQR 7 to 10), and mean initial surgery-measured systolic BP value of 139.4 mm Hg (SD 16.0). The patient characteristics of the comparator group were as shown in Table 1. Therefore, the difference in demographics between telemonitoring and control patients were more pronounced than for the overall cohort, except average initial systolic BP value, which was very similar between groups.

Mixed effects models were fitted to the outcomes to compare telemonitored patients with the comparator group, adjusting for SIMD category (<5 or ≥5), age, sex, and initial systolic BP (see S4 Table). Time spent in all appointments in the year was significantly reduced in the telemonitoring group (adjusted mean difference 16.1 minutes, 95% CI 0.1 to 32.1 minutes, *p* = 0.048). Although the reduction in time spent in face-to-face appointments did not achieve statistical significance, the observed reduction was clinically relevant (12.7 minutes, 95% CI −0.5 to 25.9 minutes, *p* = 0.059). Note that confidence intervals were wide due to the relatively small sample size in the telemonitoring group. There was no significant difference in total number of appointments (face-to-face or overall) in the adjusted analysis.

## Change in BP over time

During the study period, there were 399 patients in the 8 evaluation practices who had at least 1 year of follow-up and 2 telemonitored BPs at least 6 months apart. This group had similar initial

**Table 4. Patient characteristics in the telemonitoring and comparator groups who had at least 1 year of follow-up and 2 blood pressure readings at least 6 months apart.**

| Characteristic | Telemonitoring group ($n$ = 399*) | Comparator group ($n$ = 3,484*) |
|---|---|---|
| Female sex | 182/399 (46%) | 1,845/3,484 (53%) |
| Age (years) | Median 64 (IQR 56 to 70, min 29, max 89) | Median 71 (IQR 62 to 79, min 20, max 90) |
| SIMD 2012 decile | Median 9 (IQR 6 to 10, min 2, max 10) | Median 7 (IQR 5 to 10, min 1, max 10) |
| Initial systolic blood pressure reading in the surgery (after September 2015) | Median 138 (IQR 128 to 150, min 100, max 188) | Median 138 (IQR 130 to 150, min 71, max 240) |

*SIMD decile and systolic blood pressure were based on $n$ = 392 in the telemonitoring group. SIMD decile was based on $n$ = 3,436 in the comparator group.

max, maximum; min, minimum; SIMD, Scottish Index of Multiple Deprivation.

surgery-measured systolic BP readings compared to other hypertension patients in the practices who had not been telemonitored (who also had at least 2 BP readings at least 6 months apart). However, there was a lower percentage of females in the telemonitoring group than in the comparator group, patients were younger on average, and had a higher median SIMD decile on average (median 9 versus 7), indicating a more affluent population (see Table 4).

The median difference between the second telemonitored BP and the last was 6 mm Hg (IQR −3 to 15) for systolic BP and 4 mm Hg (IQR −1 to 10) for diastolic BP. Decreases in BP were greater for those whose BP was initially uncontrolled: People whose systolic BP was ≥135 mm Hg had a median reduction in BP of 13 mm Hg (IQR 6 to 23), whereas people whose BP was <135 had no change in the median BP (IQR −7 to 7). This reduction in systolic BP was consistent across age, sex, and deprivation (see S5 Table), and was reflected in the proportion achieving control of hypertension (see Table 5).

Fig 2 shows how mean BP changed over time for 185 people who recorded at least 1 BP reading per month for 12 months after the start of telemonitoring. The relative variability in the mean between groups and over time is represented in the plot by vertical error bars (±1 standard error). A similar plot based on the full $n$ = 399 is shown in S8 Fig, which shows a very similar relationship over time. (Note that the data are not completely paired in S8 Fig as some patients did not record monthly readings continuously and so the points are not joined together).

## Clinician responsiveness to high BP readings

We searched the Docman reports in the 8 practices for instances where BP control was poor and would normally result in management changes. We found 44 instances of average systolic BP > 160 mm Hg and 3 instances of an average systolic BP < 90 mm Hg in Docman reports.

**Table 5. Number (percent) of patient participants with various levels of uncontrolled hypertension at baseline and follow-up.**

| Blood pressure reading | Second telemonitored reading | Last telemonitored reading (6–12 months later) |
|---|---|---|
| SBP ≥ 135 mm Hg | 190/399 (48%) | 94/399 (24%) |
| SBP ≥ 140 mm Hg | 138/399 (35%) | 51/399 (13%) |
| SBP ≥ 145 mm Hg | 92/399 (23%) | 37/399 (9%) |
| SBP ≥ 150 mm Hg | 62/399 (16%) | 20/399 (5%) |
| DBP ≥ 85 mm Hg | 138/399 (35%) | 66/399 (17%) |
| DBP ≥ 90 mm Hg | 90/399 (23%) | 23/399 (6%) |

DBP, diastolic blood pressure; SBP, systolic blood pressure.

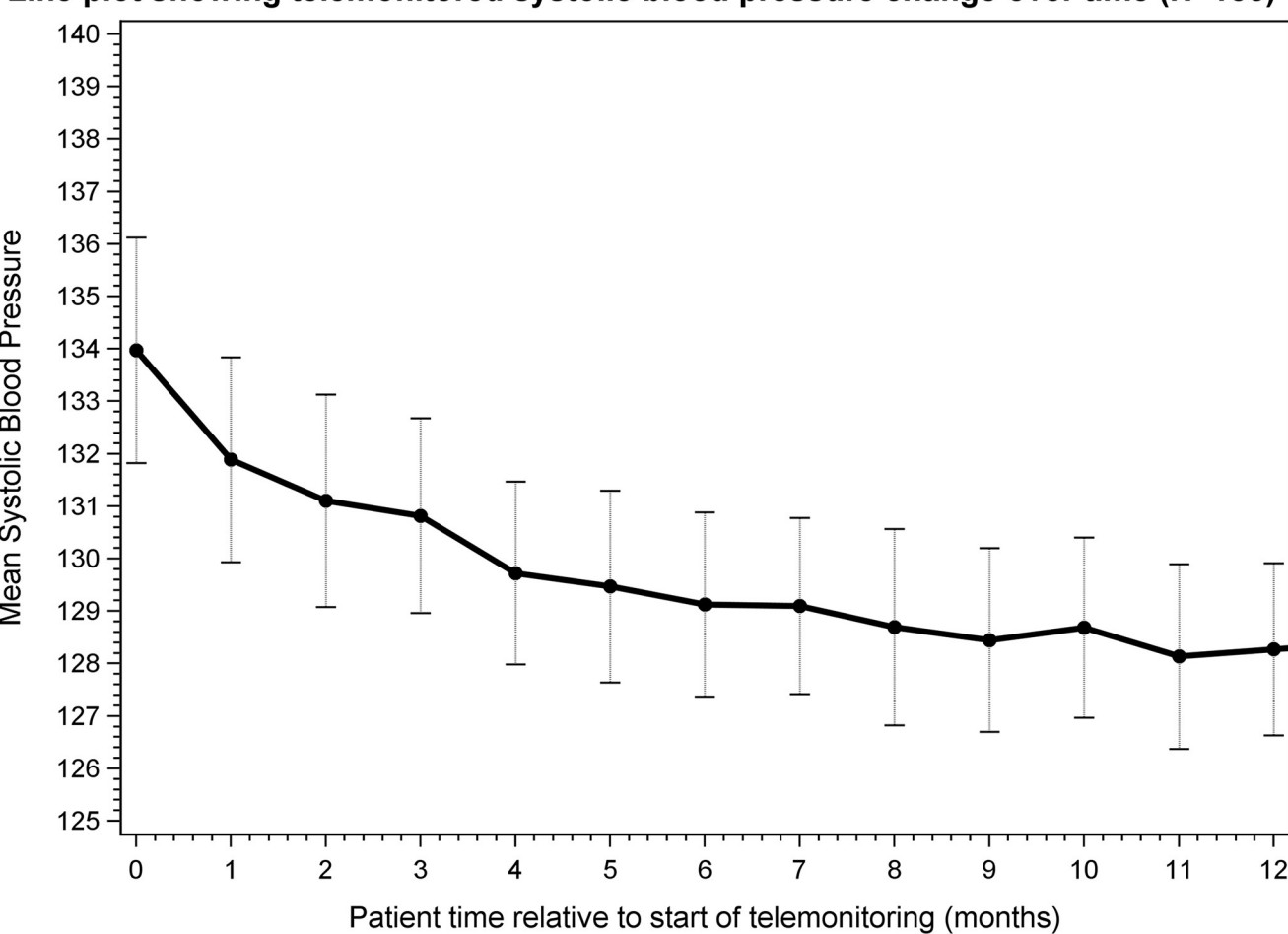

**Fig 2. Line plot showing systolic blood pressure change over time in 185 people who had at least 1 blood pressure reading per month for 12 months after the start of telemonitoring.**

The median number of days from first systolic BP > 160 mm Hg or <90 mm Hg appearing in the medical record (transcribed from the Docman report) to clinical review and action (or not) was 13 (range 0–91 days). Actions recorded in the primary care records occurring in response to these reports are summarised in S6 Table.

## Change in number of anti-hypertensive medications prescribed

For 622 patients taking part in Scale-Up BP for 1 year, we calculated average DDD of BP-lowering medications in the 45 days before they started telemonitoring and, to allow for discovery of a raised BP and time to respond to it, for a similar period 4–6 months after they started telemonitoring. Average DDD rose from 2.08 to 2.35 between those timepoints, a rise of 12% (See S7 Table). Additionally, there was a small increase in the total number of prescriptions issued for anti-hypertensive medications to patients over the whole year (1.06 [SD 4.77] additional prescriptions). Prescription numbers increased more in people whose baseline BP was >135 mm Hg and who were from more deprived areas. Similar changes were found in patients with raised BP not taking part in Scale-Up BP (see S8 Table).

## Perceptions of the implementation

Details of the process evaluation including observation of the implementation process and interviews with clinicians and patients can be found in S3 Text, S9 and S10 Tables. In summary, clinicians found that getting regular reports integrated with their usual data handling practices was particularly helpful. Continued support from the implementation team, the involvement of local champions, and patient enthusiasm for the service were all instrumental in building confidence in the process. Initially, starting patients on the system was seen as time consuming, but this improved with time and was perceived as balanced by subsequent time saving. Practice teams adapted aspects of the intervention, particularly data handling, to their own routines.

# Discussion

## Summary of findings

This study shows that a telemonitoring system for BP monitoring using software to integrate it with normal primary care work patterns can be implemented at scale. BP control improved, in line with results found in RCTs of telemonitoring, probably mediated by an intensification in therapy [16,33–36]. This new model of care was associated with an observed reduction in the number of face-to-face appointments and consulting time. The well-recognised barriers to implementation of new technologies (lack of confidence in technology, workload fears, lack of time to learn and introduce new things, and scepticism that the implementation will improve patient care or efficiency) were overcome through engaging frontline clinicians in the development of the system, particularly local champions, and strong continuous support from a facilitator team. While it was relatively straightforward to persuade practices of the likely benefits of the intervention and to undertake training, they varied in translating that training into action. Some practices required several training sessions before they started regular recruitment, while some did not get started at all, usually citing lack of time. Others, however, recruited large numbers of patients to the project, and their success persuaded other practices to follow them. Patients liked the system, and relatively few discontinued; however, it is concerning that people with less well controlled BP were overrepresented among those who discontinued, and this requires further investigation.

## Strengths and limitations

This intervention was implemented in typical primary care practices by clinical staff for patients for routine care purposes rather than in a research setting. Integration with practice data management routines and the ability to adapt the intervention encouraged uptake by practices and, apart from initial training and remote technical support, needed little additional support. The use of routine data sources means that any evaluation can be continued in future to determine the longer-term effects of the intervention.

The Scale-Up BP patients in the 8 evaluation practices were self-selected or chosen by clinical staff; they were slightly less deprived and younger on average than others in the practices with hypertension and may have been more able to participate in terms of self-management, or more motivated. Nonetheless, older and more deprived participating patients benefitted equally or more from the intervention. A component of the large reduction in BP (15 mm Hg) over time for patients who initially had uncontrolled hypertension could be considered as regression to the mean, but this was a largely unselected group in terms of BP control, and the reduction in overall mean is unlikely therefore to be a totally random effect. However, the purpose of our study was not to prove effectiveness (as this has already been established) [13] but

to show that telemonitoring can be implemented at scale. It is reassuring that the reduction in mean systolic BP was similar to or exceeded those of the intervention groups in UK RCTs with similar patient characteristics and that showed significant reductions in BP compared to a control group [16,33–36] (see S11 Table).

Patients were texting in their BP readings, and this could have been a source of bias. However, our previous work [37] showed that end-digit and target preference by patients, if it occurred, was relatively trivial and considerably less than is found in office measurements [38]. Although overall contact time and face-to-face appointments fell both in absolute terms and in comparison with people with hypertension who were not participating in telemonitoring, time spent dealing with abnormal reports in an ad hoc way that did not involve the appointment system was harder to capture. However, practices with large numbers of patients on the telemonitoring system believed that it was saving them time. Additionally, we did not capture the time spent by clinicians in the evening recruitment meeting. However, these lasted around 90 minutes, involved 2 clinicians, and typically recruited around 70 patients, so were very time efficient.

The sample size of telemonitoring patients with valid resource use data was quite small ($n = 118$) in comparison to the $n = 430$ patients in the telemonitoring group with a full year of follow-up, perhaps reflecting difficulties in capturing resource use data. Caution is therefore needed when seeking to generalise the descriptive data to a wider population. Our use of a control group was important in this situation so that we had valid comparators whose data were collected under the same conditions.

## Discussion in relation to published literature

Previous studies of telemonitoring have been in the context of RCTs with practices contributing relatively small numbers of patients that are relatively easy to manage. In this study some practices were recruiting hundreds of patients. Systematic reviews of telemonitoring of hypertension suggest that it is effective [15] in reducing BP, but that the effectiveness depends on the intensity of the intervention [13]. The most effective systems were similar to Scale-Up BP. Methods of assessing change in BP vary from study to study, but those using research nurse clinic measurements demonstrated reductions similar to those in Scale-Up BP for patients whose baseline BP was uncontrolled. There have been no direct trials to our knowledge exploring the impact of telemonitoring on cardiovascular outcomes, but based on previous studies of anti-hypertensive agents, BP reductions of the magnitude achieved in this study, probably through intensification of anti-hypertensive therapy, if sustained, would be expected to lead to a greater than 15% reduction in risk of stroke and a greater than 10% reduction in risk of coronary heart disease [39].

## Implications for policy, practice, and future research

Telemonitoring of hypertension has been shown to be cost-effective [14,15]. Sphygmomanometers have become less expensive in recent years and in Lothian will in future be provided by the Health Board. Low-cost telemonitoring systems have become available, potentially lowering costs further. We believe that the integrated system we tested will improve efficiency. The system has recently been adopted by the Scottish Government, with plans to roll it out across the country. Further work is underway to determine the optimum strategies to enrol patients in the system and to get them to adhere to it. Further integration with the primary care medical record and making the reports available to secondary care is planned, as is the development of an app-based solution and improved asynchronous communication with patients. An economic evaluation will be the subject of a future paper.

## Conclusion

Scale-Up BP has demonstrated that improvements in BP control similar to those found in RCTs of telemonitoring in hypertension can be achieved when implemented in routine practice and at scale and that this is accomplished with no increase in workload. The strategy of integrating the telemonitoring system with current data handling routines was critical in the adoption and is a model for managing other long-term conditions. Based on the findings from this implementation study, we conclude that Scale-up BP is ready for routine use across NHS Scotland and possibly also other parts of the UK.

## Supporting information

**S1 Fig. The Scale-UP BP system.**
(TIFF)

**S2 Fig. Report sent to GPs at monthly, 3-monthly, or 6-monthly frequency as required.**
(TIF)

**S3 Fig. Scale-Up recruitment over time.**
(TIF)

**S4 Fig. The range in the number of readings submitted by Scale-UP BP participants.**
(TIF)

**S5 Fig. The probability of dropping out of Scale-Up BP varied with baseline BP.**
(TIF)

**S6 Fig. Line plot showing systolic BP change over time in all $n$ = 399 patients who had at least 2 telemonitored BP readings 6–12 months apart.**
(TIF)

**S7 Fig. The observed increase in appointments just prior to the start of telemonitoring.**
(TIF)

**S8 Fig. Line plot showing telemonitored BP over time.**
(TIF)

**S1 STARI Checklist.**
(DOCX)

**S1 Table. Barriers identified to implementing telemonitoring at scale, and proposed solutions.**
(DOCX)

**S2 Table. Logistic regression results for the analysis of attrition.**
(DOCX)

**S3 Table. Number of all appointment activities and face-to-face appointments in telemonitored group and comparator group, by age, sex, deprivation status, and baseline BP.**
Excludes 2 weeks either side of the anchor point.
(DOCX)

**S4 Table. Adjusted mean differences of resource use outcomes for telemonitoring versus comparator group from linear mixed effects models, adjusting for GP practice (as a random effect), sex, initial systolic BP (SBP > 145, SBP 135–145, SBP < 135), and age (<65**

versus 65+ years) (*n* = 7,429).
(DOCX)

**S5 Table. Summary statistics for the reduction between the second and last reading within 12 months in systolic BP stratified by age, sex, SIMD [1], and baseline BP.**
(DOCX)

**S6 Table. Clinician actions following a patient recording of systolic BP < 90 mm Hg or >160 mm Hg.**
(DOCX)

**S7 Table. The change in defined daily dose of BP-lowering drugs in the telemonitored group from before to after commencing telemonitoring.**
(DOCX)

**S8 Table. Number of prescriptions per patient in year before compared to year after anchor point, overall and stratified by age, sex, SIMD, and initial BP in intervention and comparator groups.**
(DOCX)

**S9 Table. Relevant quotations from interviews categorised according to themes.**
(DOCX)

**S10 Table. Barriers and facilitators identified during the evaluation phase and potential solutions suggested by participants.**
(DOCX)

**S11 Table. Change in BP in intervention groups of UK telemonitoring trials comparable to Scale-Up BP participants.**
(DOCX)

**S1 Text. The Scale-Up BP intervention.**
(DOCX)

**S2 Text. Methods for qualitative exploration of patient and professional perceptions.**
(DOCX)

**S3 Text. Perceptions of the implementation of Scale-UP BP.**
(DOCX)

## Acknowledgments

We would like to thank the patients and practice staff who took part, and Grahame Cumming, Elizabeth Payne, Alison McAulay, Arek Makarenko, and Daniel Plenderleith of NHS Lothian. We acknowledge the support of NHS Research Scotland Primary Care Network and Lothian Safe Haven, particularly Allan Walker. Special thanks to Margaret Whoriskey and Michelle Brogan of Scottish Government Technology Enabled Care and Richard Forsyth of the British Heart Foundation.

## Author Contributions

**Conceptualization:** Janet Hanley, Hilary Pinnock, Paul Padfield, Brian McKinstry.

**Formal analysis:** Vicky Hammersley, Richard Parker, Mary Paterson, Andrew Stoddart, Hyeon Gyeong Park, Brian McKinstry.

**Funding acquisition:** Richard Parker, Janet Hanley, Hilary Pinnock, Paul Padfield, Andrew Stoddart, Brian McKinstry.

**Investigation:** Vicky Hammersley, Mary Paterson.

**Methodology:** Richard Parker, Janet Hanley, Hilary Pinnock, Paul Padfield, Andrew Stoddart, Brian McKinstry.

**Project administration:** Vicky Hammersley.

**Supervision:** Janet Hanley, Aziz Sheikh, Brian McKinstry.

**Writing – original draft:** Vicky Hammersley, Brian McKinstry.

**Writing – review & editing:** Vicky Hammersley, Richard Parker, Mary Paterson, Janet Hanley, Hilary Pinnock, Paul Padfield, Andrew Stoddart, Hyeon Gyeong Park, Aziz Sheikh, Brian McKinstry.

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
