## [Decision Letter · Decision Letter 0]

11 Dec 2019

Dear Dr. McKinstry,

Thank you very much for submitting your manuscript "Telemonitoring for hypertension at scale: implementation study in routine general practice" (PMEDICINE-D-19-03237) for consideration at PLOS Medicine. 

[LINK]

In light of these reviews, I am afraid that we will not be able to accept the manuscript for publication in the journal in its current form, but we would like to consider a revised version that addresses the reviewers' and editors' comments. Obviously we cannot make any decision about publication until we have seen the revised manuscript and your response, and we plan to seek re-review by one or more of the reviewers. 

We expect to receive your revised manuscript by Jan 01 2020 11:59PM. Please email us (plosmedicine@plos.org) if you have any questions or concerns.

We look forward to receiving your revised manuscript. 

Sincerely,

Caitlin Moyer, Ph.D.

Associate Editor 

PLOS Medicine

plosmedicine.org

In general, The presentation tends to exaggerate the findings, it seems, (e.g. "results in clinically important reductions in blood pressure") please be more cautious in reporting. 

Specific points:

Title - Please revise your title according to PLOS Medicine's style. Your title must be nondeclarative and not a question. It should begin with main concept if possible. "Effect of" should be used only if causality can be inferred, i.e., for an RCT. Please place the study design ("A randomized controlled trial," "A retrospective study," "A modelling study," etc.) in the subtitle (ie, after a colon).

Abstract – there should be 3 headings ‘Background’ ‘Methods and Findings’ and ‘Conclusions’ – please revise accordingly; Please provide summary demographic details; please add a sentence on the study’s limitations as the final sentence of the ‘Methods and Findings’ section; Please avoid speculative language (with no increase (and possibly a reduction) in workload for clinicians).

Data – please provide a URL or point of contact (cant be an author) for those who do meet requirements for access – and can you please be clearer on what the ‘governance training’ is. 

At this stage, we ask that you include a short, non-technical Author Summary of your research to make findings accessible to a wide audience that includes both scientists and non-scientists. The Author Summary should immediately follow the Abstract in your revised manuscript. This text is subject to editorial change and should be distinct from the scientific abstract. Please

see our author guidelines for more information: https://journals.plos.org/plosmedicine/s/revising-your-manuscript#loc-author-summary

Please remove data sharing / funding etc from main text (page 21) as these will be pulled in from the EM submission system, from meta data. 

I wonder if it would be helpful to include a table of participant characteristics?

Checklist – please report using sections and paragraphs instead of page numbers as these can change through formatting / revisions, etc. 

Comments from the reviewers:

Reviewer #1: See attachment

Michael Dewey

Reviewer #2: This is an observational study of the application at-scale of telemonitoring for hypertension in Scotland. The authors suggest that telemonitoring improved BP control (not very convincing) and did not increase consultation time in primary care (fairly convincing). Ultimately, this study achieves its primary aim (feasibility) but does not convincingly show that this improves BP control or saves GP time. It provides no evidence of improved outcome. In summary, it shows that telemonitoring of hypertension at scale can be done but does not show that it should be done. Nor do previous trials show convincing evidence of benefit. Nonetheless, as a system of care it does SEEM both rational and feasible.

Abstract

This is VERY misleading. The abstract should better reflect what's in the paper.

Background: what evidence is there that telemonitoring improves the outcome of hypertension (which most people will take to mean reductions in stroke and MI and an increase in longevity)?

Results: this mentions 3,200 patients but tables show at most 905 participating in telemonitoring. This seems dishonest even if correct on a technicality. 

Introduction

The introduction is marred by statistics that have almost certainly been cut and pasted from another paper that has not adequately checked its facts. Assuming that there are 7 billion people in the world, and that average life expectancy is no greater than age 70 years (optimistic at present), then there must be 100 million deaths per year. So, hypertension is related to less than 10% of deaths? Possible but is it really this low? I also doubt that 2.8 billion people have hypertension (although accept that more than 2.8 billion may live to develop it) even if this is stated by WHO. What proportion of the world population is aged >50 years - age being the single most important determinant of hypertension. This section should be deleted. Why not just say that "hypertension is common in the people aged >50 years and associated with a substantial increase in morbidity and mortality"? This would be both brief and accurate.

The subsequent three paragraphs are excellent! Why cast doubt on them by the fake-news in the first?

Methods

Well described generally, but why was SBP 135mmHg chosen and not 130 or 140mmHg?

The investigators have been cautious in choosing their baseline BP reading (good!)

Results

Please report data to a sensible number of decimal points (for BP this will be whole numbers - not two decimal places!). 

Median IQR of patients per practice should be shown. Median/IQR for age and other descriptive variables is more informative for the clinical reader than mean/SD. Also, I am not sure that reporting mean change with a SD that is 10 times greater is correct or appropriate. I would find it much easier to understand median/IQR.

Patient population: female and male are terms better reserved for victims, criminals and rats? How about talking about men and women? 

The reduction in SBP is what might be expected in any observational study or in the placebo-group of a clinical trial, although avoiding using the first reading partially offsets this criticism.

Number of recordings might be better expressed as a rate per month for patients in years one, two and three to see if enthusiasm changes with time. Also, looks like some people are anxious (many BP recorded) - did those with a high frequency of readings have higher BP and did it lead to more effective therapy / greater decline in BP? Figure S4 is helpful.

The demonstration that telemonitoring did not increase consultation rates is reassuring.

The authors should avoid terms such as "SIMD<5" which might mean something in Scotland but nowhere else! <5 should mean less deprivation but curiously in Scotland less is more (deprivation - weird!).

Figure 2. I suspect this is misleading as it implies that all patients had readings at all times (how else could the dots be connected and how else could they all have the same baseline value?). This Figure needs to be corrected showing only paired data between baseline and each time-point and NOT connecting every dot.

Page 16: Is 160mmHg in a patient with known hypertension a matter of urgency for a patient or GP? I don't think so. Where is the evidence?

Page 17: Please don't refer to medicines as drugs. "Drugs" is a derogatory or slang term that is ambiguous in its meaning.

Why would you expect an increase in prescriptions for patients with adequately controlled BP?

Please report changes in medications as median/IQR.

Conclusions: these should specify the "benefits" observed in RCTs, which are, as far as I can see, better BP control and not improved outcome (which we might hope for - but fact should be distinguished from hope).

Tables 1 and 2

These need to be revised to show BP to whole numbers (who treats to 0.01mmHg?)

Layout should be much clearer - perhaps using only median/IQR data.

Reviewer #3: This study in treated hypertensives in Scotland assessed the feasibility of implementing home-BP telemonitoring and its impact in primary care. The authors concluded that home BP telemonitoring can be implemented in treated hypertensives in primary care and improves BP control together with possible reduction in clinicians' workload. 

COMMENTS

1. Data to show that 8 practices with detailed data represent the total 126?

2. Not sure whether it is clear how the control group was selected.

3. Page 12: 'In the eight evaluation practices, 905 patients signed up to use the telemonitoring system': Were consecutive patients invited to use the home BP telemonitoring system, or only selected ones? How many patients refused to participate?

4. Page 10: The difference between the second and the last BP reading occurring 6-12 months after the second reading was calculated for each patient: I think, due to the inherent BP variability, a single reading should not be used to represent home BP at time points. A major advantage of home BP is the large number of readings obtained. More than one home BP readings should be used to represent each time point.

5. The potential impact of 'evening group meetings' has to be taken into. It requires additional resources and costs and might have contributed to the success of home BP telemonitoring.

6. It is important to realize that the average number of readings (42 per year) is based on data from 430 patients.

7. The fact that patients with SBP of 135 mmHg or above had over twice the odds of discontinuing compared to patients with SBP below 135 mmHg is a problem, as the former are those who need closer monitoring - not the latter.

8. Reduction in appointments: Did patients with reduced number of visits have low home BP? If not, then this is not necessarily a good sign. Is this what the supplementary table S3 says?

9. Page 12: 28 (IQR 14 to 76) BP readings were submitted during this period - Is this 28 readings in 2,5 years? Please make clearer. 

10. Page 12: 'by July 2019 75/126 practices in Lothian had participated' and 3200 patients by July 2019: Better say how many months after project initiation.

11. Change in number of drugs prescribed: DDDs rose from 2.08 to 2.35 between those times, a rise of 12%. - What is the net effect of home BP telemonitoring on this finding?

12. Table 2: N=118 is a relatively small sample to extrapolate to total 3.200 or to compare vs 9.061.

13. Figure 2 should show a comparator group.

14. No need to provide 2 decimal digits in mmHg of BP.

[LINK]

---

## [Decision Letter · Decision Letter 1]

8 Apr 2020

Dear Dr. McKinstry,

Thank you very much for re-submitting your manuscript "Telemonitoring at scale for hypertension in primary care: an implementation study" (PMEDICINE-D-19-03237R1) for review by PLOS Medicine.

I have discussed the paper with my colleagues and the academic editor and it was also seen again by 2 of the previous reviewers. I am pleased to say that provided the remaining editorial and production issues are dealt with we are planning to accept the paper for publication in the journal.

[LINK]

We look forward to receiving the revised manuscript by Apr 15 2020 11:59PM. 

Sincerely,

Clare Stone PhD

Acting Editor-in-Chief 

PLOS Medicine

plosmedicine.org

Requests from Editors:

We received comments from the Academic Editor that this study, with 8 GP practices do not constitute at scale and we therefore ask that this be amended in the title and elsewhere (such as line 499)

So, for example we would suggest the title be changed to: "Telemonitoring of hypertension in GP practices in Scotland; an implementation study”

Line 3 – ‘Summary’ should be ‘Abstract’

Line 51 add ‘ Author Summary’

Comments from Reviewers:

Reviewer #1: The authors have addressed my points.

Michael Dewey

Reviewer #3: No further comments

[LINK]

---

## [Editor Report · Decision Letter 2]

22 May 2020

Dear Prof McKinstry, 

On behalf of my colleagues and the academic editor, Dr. Kazem Rahimi, I am delighted to inform you that your manuscript entitled "Telemonitoring for hypertension in primary care: an implementation study" (PMEDICINE-D-19-03237R2) has been accepted for publication in PLOS Medicine. 

PRODUCTION PROCESS

PRESS

PROFILE INFORMATION

Thank you again for submitting the manuscript to PLOS Medicine. We look forward to publishing it. 

Best wishes, 

Caitlin Moyer, Ph.D.

Associate Editor 

PLOS Medicine

plosmedicine.org